# Ethnomedicinal appraisal of plants used for the treatment of gastrointestinal complaints by tribal communities living in Diamir district, Western Himalayas, Pakistan

Rahmat Wali[1]*, Muhammad Faraz Khan[1,2]*, Ansar Mahmood[2], Majid Mahmood[3], Rahmatullah Qureshi[1], Khawaja Shafique Ahmad[2], Zia-ur-Rehman Mashwani[1]

1 Department of Botany, PMAS Arid Agriculture University Rawalpindi, Rawalpindi, Pakistan, 2 Department of Botany, Faculty of Basic and Applied Sciences, University of Poonch, Rawalakot, AJ&K, Pakistan, 3 Department of Zoology, Faculty of Basic and Applied Sciences, University of Poonch, Rawalakot, AJ&K, Pakistan

* mfarazkhan@upr.edu.pk (MFK); rahmatwali80@gmail.com (RW)

**Data Availability Statement:** All relevant data are within the paper and its Supporting Information files.

## Abstract

Majority of the mountain dwelling communities living in the Himalayas rely on traditional herbal medicines for primary healthcare needs. Present study was conducted in fairy meadows and allied valleys in District Diamir, Gilgit Baltistan autonomous territory in northern Pakistan. Documentation of traditional medicinal knowledge (TMK) of local communities for the treatment of gastrointestinal disorders was carried out as a component of a wider medico-botanical expedition conducted in the entire base camp of the great Nanga Parbat peak during 2016–19. Various ethnobotanical parameters i.e. use value (UV), informant consensus factor (ICF), Fidelity level (FL), direct matrix ranking test (DMRT) and preference ranking (PR) were applied to evaluate the data collected during field surveys. The plants were also subjected to a comparative review for novelty assessment. A total of 61 medicinal plant species belonging to 55 genera and 35 families are reported here for the treatment of GIDs. Compositae was the leading family with 8 (13%) species. Fourteen gastrointestinal disorders were cured with 32% taxon were reported for stomachic followed by diarrhea (15%) and constipation (14%). Highest use reports (5) and use citations (207) were reported for *Mentha longifolia* L. while highest UV (1.79) was obtained for *Artemisia maritima* L. *Hylotelephium telephioides* (Ledeb.), *A. maritima*, *M. longifolia*, *M. piperita* L., *Allium cepa* L., and *A. annua* L. exhibited 100% FL. Highest ICF was calculated against dysentery and flatulence. DMRT ranked *Prunus persica* L. first for its multipurpose uses. Taking constipation as a reference gastrointestinal disease, PR for ten plant species was calculated where *H. telephioides* was ranked first followed by *A. maritima*. Present study concluded that 19 out of 61 plant species were documented for the first time with novel medicinal uses to cure GIDs. These plant species could act as potential reservoirs of novel lead compounds for the treatments of gastrointestinal disorders.

**Funding:** The author(s) received no specific funding for this work

## Introduction

More than 70% of rural population in developing countries relies on traditional medicinal plants for their primary health needs [1]. Digestive track ailments, collectively known as gastrointestinal disorders (GIDs) are cited more frequently in mountain dwelling communities where poor sanitation practices are more common [2]. GIDs range from short-term indigestion or flatulence to long-term structural anomalies and chronic illnesses having a substantial influence on morbidity and mortality. According to world health organization (WHO) estimates, GIDs, caused nearly 1 million adult deaths worldwide during 2019 where diarrhea alone was responsible for 370,000 death in children under the age of 5 years [3]. Apart from mortality and morbidity, GIDs have some intrinsic relationship with overall human wellbeing. Recent data has suggested that some GIDs act as major contributing factor in onset of neurological disorders such as Parkinson's Disease and Alzheimer's Disease [4]. Botanicals formulations and plant extracts are recently being investigated as modulators for "brain-gut-microbiota axis" a recently adopted phrase for emerging interface of human ailments [5, 6]. Use of medicinal plants for GIDs has extensively been reported from ethnobotanical expedition across the world since few decades [7–12]. Field surveys have reported that instant relief from gut ailments strengthens the trust of locals on plant derived home remedies as alternative to costly and slowly acting allopathic drugs [13, 14]. Moreover, many of the GIDs are considered taboos in conservative Alpine Asian culture. Bad breath, Piles, irritable bowel syndrome, gas and many other GIDs are never shared comfortably with a physician, who, in most of the cases would be a non-native person [15, 16]. Either way, the trust of indigenous people on traditional medicine used for GIDs is exceptionally high. On the other hand, plant-based therapies are the only choice for rural communities in various remote areas of third world countries including Pakistan. Tribal communities such as those living in vales and river terraces of Karakorum and Himalayas conserve a very unique assemblage of plant use practices [17]. Gilgi-Baltistan along with adjoining Chinese areas represent a greater biodiversity hotspot hosting more than 300 medicinal and aromatic plants [18, 19]. Recently, a couple of targeted medico-botanical expeditions have revealed various medicinally important species of genus Berberis are documented with extensive use practices [20, 21]. Owing to ethnic diversity, and historical linkages with various civilizations, entire region of Gilgit-Baltistan is very rich in traditional medicinal practices [22].For centuries, the area has acted as a pedestrian crossroad between mainland China, Central Asia, and Indian subcontinent. This has led to a blending of three greater civilizations [23–25]. That is why ancient ayurvedic wisdom shares many commonalities with Traditional Chinese medicine (TCM) [26]. There are certain geographic limitations that impose a sustainable food-cum-medicine use strategy of indigenous flora as an imperative of survival in almost all areas of upper Indus basin (UIB) including District Diamer [27].

The study area comprises of the "Fairy Meadows" and allied tribal setting at the base camp of Nanga Parbat, the second highest peak in Pakistan. Fairy Meadows, locally known as "Joot", was named by German mountaineers who got mesmerized on their maiden journey to the areas. It was declared a national park in 1995 as an acknowledgement of diverse flora, fauna, and geographic features. There is a large number of endemic species though not documented very well for their taxonomic status and medicinal uses [28]. The road leading to study areas is declared as world's second most dangerous track; that speaks of its remoteness and staying largely unexplored yet. More than 15 km of jeep ride is required for mere entry in to the 'Fairy Meadows' while rest of the study sites lie on small river-let basins and ascending terraces towards south-western and northern faces of Nanga Parbat. Health infrastructure is next to

nothing in the area that implies almost complete reliance of the inhabitants of locally available healing ways largely comprised of plant derived home recipes or formulations prescribed by locally practicing herbal healers.

## Materials and methods

### Statement of ethics

The field study was conducted in strict compliance to ethical standards for ethno botanical field studies. Formal ethical approval was acquired from institutional ethics committee vide.

### Filed surveys

After preliminary survey, a series of target expeditions were conducted in different villages and seasonal settlement across alpine landscape during summers. A detailed inventory of medicinal plants was developed during April 2018 to July 2019. Structured and semi-structured interviews were conducted for the ethnobotanical data collection. Informant consent and consent for the publication was taken from the elders of the community before the interviews.

### Study area

District Diamir shares boundaries with Khyber Pakhtunkhwa (KPK) province in the south, Azad Kashmir in the west, Ghizer District in the east and Astore district in the north. The study area has small villages, having human settlements namely Buner Das, Halaly, Thamros, Gashot, Pakora and some small patches with human settlements. Extreme summer temperature rises up to 38°C and winter temperature fall below 0°C in the valleys [29]. The Nanga Parbat (ninth-highest peak on the earth measuring 8126 m ASL) is the western anchor of Himalayan ranges known as the last eight thousander in the mighty mountain range. The study sites are situated between 35°23'14.3"N latitude and 74°34'44.7"E longitude, at an elevation range between 934–7937 m above the sea level (**Fig 1**). District Diamir covers an area of 10936 Km$^2$, divided into two tehsils which are: tehsil Chilas and tehsil Darel/Tangir. Gilgit Baltistan has an important strategic location as it hosts the confluence of three great world mountainous ranges, the Karakorum, the Hindu Kush and the Himalayas, providing the habitat for at least 10% of global flora [30].

### Participants of the study area

The study area has a rich diversity of culture and ethnic groups. Different languages like Shina, Gojri, Kashmiri and Dulasgariya are spoken in the area. Shina is the dominant language as 100% of the studied population can speak and understand it. Various ethnic groups like Sheen, Youshkon, Gujar, Dulasgar and Kashmiri reside in the study area. A total of 166 individuals including 115 men and 51women were interviewed about the use of selected medicinal plants for the treatment of GIDs (Table 1).

### Documentation of ethnomedicinal data

Ethnomedicinal data was documented using questioners (structured and semi-structured) interviews, participant observations, guided field walks, and focus group discussions [31–33]. Information like local name of the plant, plant parts used, recipe, disease treated, and other information were collected from locals. In addition to this, standard data collection methods [34] have been followed for the documentation of indigenous knowledge of community on health, use, conservation and threats of medicinal plants. Prior rural consent and consent for the publication was obtained from the elders (Lumberdar) orally and the ethical standards of

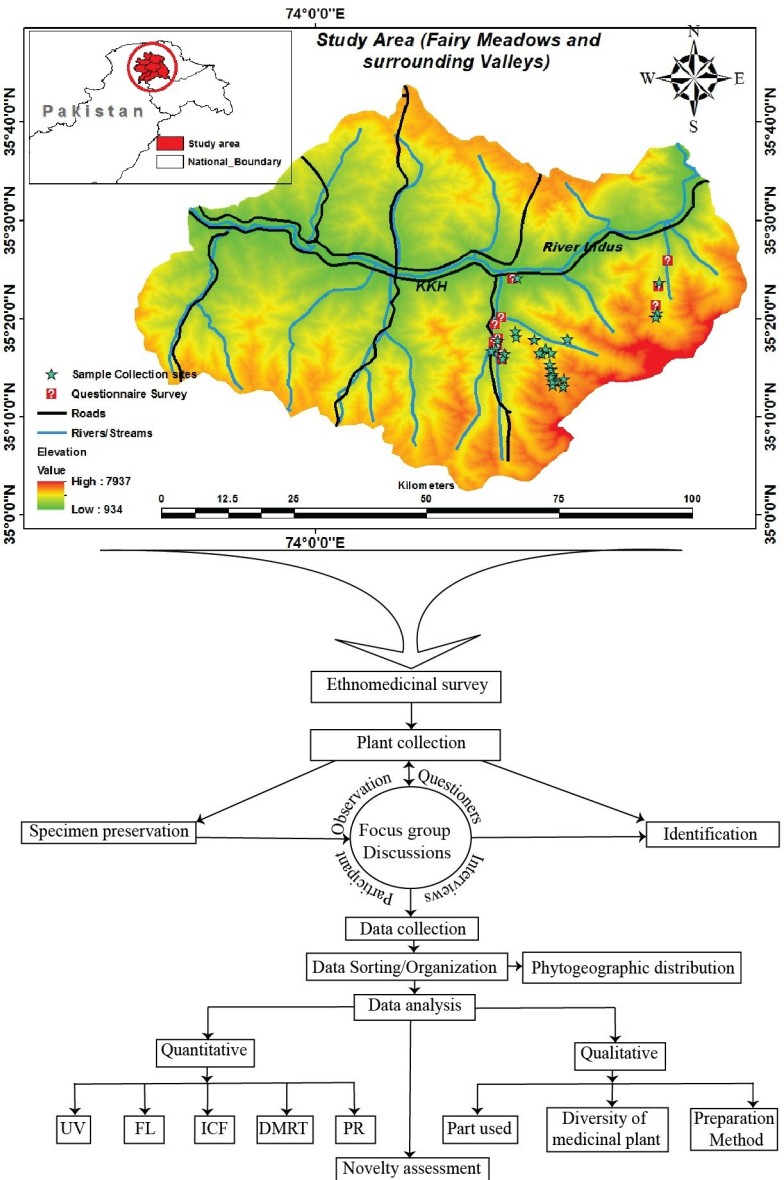

**Fig 1. Map of the study area with schematic work flow diagram.**

the Society for Economic Botany and International Society of Ethnobiology [35] were followed. All the interviews were conducted in local Shina language.

## Plant collection and identification

Following the interviews and discussions, voucher specimens were collected. These specimens were pressed and mounted on herbarium sheets. Botanical identification of the specimens was done by Prof. Dr. Rahmatullah Qureshi, at Pir Mehr Ali Shah University of Arid Agriculture Rawalpindi (PMAS UAAR) Pakistan, and authenticated with the help of flora of Pakistan (http://www.efloras.org/flora_page.aspx?flora_id=5), The Plant List (http://www.theplantlist.org/), International Plant Name Index (https://www.ipni.org/) and World Flora Online (http://www.worldfloraonline.org/) with already identified specimens. Air-dried specimens

**Table 1. Demographics of the study area.**

| Demographic feature | Criteria | Number of informants | Percentage |
|---|---|---|---|
| Gender of informants | Male | 115 | 69.3 |
| | Female | 51 | 30.7 |
| Distribution by Village | Buner Das | 27 | 16.3 |
| | Thamros | 40 | 24.1 |
| | Gashot | 36 | 21.7 |
| | Pakora | 30 | 18.1 |
| | Keli Jail | 15 | 9 |
| | Halaly | 18 | 10.8 |
| Age of Informants | Between 20 to 30 | 16 | 9.6 |
| | Between 31 to 40 | 23 | 13.9 |
| | Between 41 to 50 | 39 | 23.5 |
| | Between 51 to 60 | 45 | 27.1 |
| | Above 60 | 43 | 25.9 |
| Marital status | Married | 123 | 74.1 |
| | Unmarried | 36 | 21.7 |
| | Widow | 7 | 4.2 |
| Education level | Illiterate | 97 | 58.4 |
| | Elementary School | 25 | 15.1 |
| | Secondary School | 22 | 13.3 |
| | College | 15 | 9 |
| | University | 7 | 4.2 |
| Social Livelihood | Farmer | 112 | 67.5 |
| | Retired | 5 | 3 |
| | Shepherd | 31 | 18.7 |
| | Others | 18 | 10.8 |
| Residence | Village | 155 | 93.4 |
| | Seasonal migrants | 11 | 6.6 |
| Ethnic group | Youshkon | 79 | 47.6 |
| | Sheen | 34 | 20.5 |
| | Gujar | 38 | 22.9 |
| | Kashmiri | 15 | 9 |
| Experience | Herbalists | 7 | 4.2 |
| | Local people | 159 | 95.8 |
| Duration of residence in the area | Less than 15 years | 9 | 5.4 |
| surveyed area | More than 15 years | 157 | 94.6 |
| Religion | Islam | 166 | 100 |

were systematically tagged, labeled and voucher number for each specimen was allotted (Table 2). The identified specimens were deposited in the herbarium of Department of Botany, PMAS UAAR for future reference.

## Data analysis

The ethnomedicinal data obtained during the field surveys was shifted to the Microsoft excel spread sheet and organized in a tabulated form for presentation. Various quantitative ethnobotanical indices like Use Value (UV), Frequency of Citation (FC), Relative Frequency of Citation (RFC), Direct Matrix Ranking (DMR), Fidelity Level percentage (FL), and Informant

**Table 2. Plant species with accession ids, GIDs treated, citations, frequency of citation (FC), relative frequency of citation (RFC) and use value (UV).**

| S. No. | Local Name | Taxonomic Name | Uses | UR | Use citations | FC | RFC | UV |
|---|---|---|---|---|---|---|---|---|
| 1 | Zoon | *Artemisia maritima* L./RW735 | Intestinal worms (65), diarrhea (43), vomiting (71) & stomachic (20) | 4 | 200 | 112 | 0.67 | 1.79 |
| 2 | Philil | *Mentha longifolia* (L.) L./RW745 | diarrhea (89), intestinal worms (41), digestive disorders (23), vomiting (44) & stomachic (10) | 5 | 207 | 140 | 0.84 | 1.48 |
| 3 | Nerlay Zoon | *Tanacetum faconeri* Hook.f./RW739 | diarrhea (89), vomiting (53), stomachic (47) | 3 | 189 | 132 | 0.80 | 1.43 |
| 4 | Philil | *Mentha piperita* L./RW736 | Diarrhea (79), indigestion (21) & ulcer (9) | 3 | 109 | 99 | 0.60 | 1.10 |
| 5 | Kasho/Paloan | *Allium cepa* L./RW744 | Diarrhea (51), vomiting (34) | 2 | 85 | 78 | 0.47 | 1.09 |
| 6 | Choro | *Pimpinella diversifolia* DC./RW746 | Stomachic (42), Intestinal worms (38) | 2 | 80 | 86 | 0.52 | 0.93 |
| 7 | Teetar | *Hylotelephium telephioides* (Ledeb.) H. Ohba/RW741 | Constipation (132) | 1 | 132 | 143 | 0.86 | 0.92 |
| 8 | Chontal | *Rheum webbianum* Wall./RW747 | Stomachic (31), constipation (18) & intestinal worms (21) | 3 | 70 | 76 | 0.46 | 0.92 |
| 9 | Nerlay Churki | *Oxyria digyna* (L.) Hill/RW740 | diarrhea (67) & digestive (36) | 2 | 103 | 112 | 0.67 | 0.92 |
| 10 | Lilio | *Viola serpens* WalL./RW748 | diarrhea (23), constipation (11) & stomachic (23) | 3 | 57 | 62 | 0.37 | 0.92 |
| 11 | Shey Lamay | Persicaria amplexicaulis (D.Don) Ronse Decr./RW749 | Stomach disorders (57) | 1 | 57 | 62 | 0.37 | 0.92 |
| 12 | Cheti Char | *Cichorium intybus* L./RW750 | Ulcer (22), stomachic (43) | 2 | 65 | 79 | 0.48 | 0.82 |
| 13 | Aaro | *Prunus persica* L./RW751 | Intestinal worms (21), stomachic (26) & gastritis (9) | 3 | 56 | 69 | 0.42 | 0.81 |
| 14 | Angrezi phang | *Ficus carica* L./RW752 | Constipation (24) & stomachic (31) | 2 | 55 | 68 | 0.41 | 0.81 |
| 15 | Joi | *Prunus armeniaca* L./RW753 | Diarrhea (32) & constipation (21) | 2 | 53 | 67 | 0.40 | 0.79 |
| 16 | Love | *Cucumis sativus* L./RW754 | Digestive (29) & purgative (30) | 2 | 59 | 77 | 0.46 | 0.77 |
| 17 | Tumurum | *Thymus serphyllum* L./RW755 | Stomachic (75) | 1 | 75 | 98 | 0.59 | 0.77 |
| 18 | Chenga | *Persicaria vivipara* (L.) Ronse Decr./RW756 | diarrhea (3) & constipation (43) | 2 | 46 | 61 | 0.37 | 0.75 |
| 19 | Goom | *Triticum aestivum* L./RW757 | Piles (56) | 1 | 56 | 76 | 0.46 | 0.74 |
| 20 | Bushi punar | *Saussurea gossypiphora* D.Don/RW742 | constipation (54) | 1 | 54 | 75 | 0.45 | 0.72 |
| 21 | Susar | *Rhododendron anthopogon* D. Don/RW758 | Stomachic (74) | 1 | 74 | 104 | 0.63 | 0.71 |
| 22 | Patrees | *Aconitum heterophyllum* Wall. ex Royle/RW759 | Stomachic (27) & intestinal worms (16) | 2 | 43 | 61 | 0.37 | 0.70 |
| 23 | Jomi | *Urtica dioica* L./RW760 | Vomiting (31) | 1 | 31 | 44 | 0.27 | 0.70 |
| 24 | Koret | *Bergenia stracheyi* (Hook.f. & Thomson) Engl./RW761 | Stomachic (47) & diarrhea (24) | 2 | 71 | 102 | 0.61 | 0.70 |
| 25 | Simbul Char | *Adiantum raddianum* C. Presl/RW762 | diarrhea (41) | 1 | 41 | 59 | 0.36 | 0.69 |
| 26 | One | *Cucurbita maxima* Duchesne/RW763 | Stomachic (27) & constipation (16) | 2 | 43 | 62 | 0.37 | 0.69 |
| 27 | Shatoo | *Ribes alpestre* Wall.ex Decne./RW764 | Ulcer (22) | 1 | 22 | 32 | 0.19 | 0.69 |
| 28 | Margosh Chontal | *Rheum australe* D. Don/RW743 | Intestinal worms (64) & constipation (32) | 2 | 96 | 142 | 0.86 | 0.68 |
| 29 | Peban Maroch | *Morus alba* L./RW765 | Stomachic (29) & constipation (21) | 2 | 50 | 74 | 0.45 | 0.68 |
| 30 | Mulo | *Raphanus sativus* L./RW766 | Stomachic (33) | 1 | 33 | 50 | 0.30 | 0.66 |
| 31 | Gizari | *Daucus carota* L./RW767 | Digestive (43) | 1 | 43 | 67 | 0.40 | 0.64 |
| 32 | Nooni Char | *Oxalis corniculata* L./RW768 | Constipation (23) & ulcer (11) | 2 | 41 | 65 | 0.39 | 0.63 |
| 33 | Chorko | *Berberis lycium* Royle/RW769 | Stomach problem (41) | 1 | 41 | 65 | 0.39 | 0.63 |
| 34 | Konay | *Echinops echinatus* Roxb./RW770 | Abdominal pain (22) | 1 | 22 | 35 | 0.21 | 0.63 |
| 35 | Churki | *Rumex hastatus* D. Don/RW771 | Stomachic (29) & flatulence (22) | 2 | 51 | 82 | 0.49 | 0.62 |
| 36 | Hailel | *Solanum nigrum* L./RW772 | Dysentery (9), stomachic (23) & ulcer (8) | 3 | 40 | 65 | 0.39 | 0.62 |
| 37 | Khakao | *Pistacia khinjuk* Stocks/RW773 | indigestion (41) | 1 | 41 | 67 | 0.40 | 0.61 |
| 38 | Khaneray Char | *Salvia* sp./RW737 | Stomachic (71) | 1 | 71 | 118 | 0.71 | 0.60 |

(*Continued*)

**Table 2.** (Continued)

| S. No. | Local Name | Taxonomic Name | Uses | UR | Use citations | FC | RFC | UV |
|---|---|---|---|---|---|---|---|---|
| 39 | Hamay | *Dysphania botrys* (L.) Mosyakin & Clemants/RW774 | Abdominal pain (6) & diarrhea (17) | 2 | 23 | 41 | 0.25 | 0.56 |
| 40 | Bhendi | *Abelmoschus esculentus* (L.) Moench/RW775 | Digestive (27) | 1 | 27 | 49 | 0.30 | 0.55 |
| 41 | Hayao | *Bunium persicum* (Boiss) B. Fedtsch./RW776 | stomachic (41) | 1 | 41 | 76 | 0.46 | 0.54 |
| 42 | Makai | *Zea mays* L./RW777 | Stomachic (32) & gastritis (27) | 2 | 59 | 111 | 0.67 | 0.53 |
| 43 | Tandur | *Datura stramonium* L./RW778 | Stomachic (32) | 1 | 32 | 61 | 0.37 | 0.52 |
| 44 | Shangali | *Cuscuta reflexa* Roxb. | Stomach disorders (23) | 1 | 23 | 44 | 0.27 | 0.52 |
| 45 | Shaftal | *Trifolium repens* L./RW737 | Stomachic (22) | 1 | 22 | 43 | 0.26 | 0.51 |
| 46 | Bangra | *Swertia petiolata* D. Don/RW738 | Stomachic (41) | 1 | 41 | 81 | 0.49 | 0.51 |
| 47 | Khapoy Patay | *Plantago himalaica* Pilg./RW779 | Stomachic (19) & diarrhea (9) | 2 | 28 | 59 | 0.36 | 0.47 |
| 48 | Chilli | *Juniperus excelsa* M.Bieb./RW780 | diarrhea (43) | 1 | 43 | 92 | 0.55 | 0.47 |
| 49 | Danoi | *Punica granatum* L./RW781 | Intestinal worms (32) | 1 | 32 | 72 | 0.43 | 0.44 |
| 50 | Zooti Ponar | *Aster himalaicus* C. B. Clarke/RW782 | Stomachic (18) | 1 | 18 | 44 | 0.27 | 0.41 |
| 51 | Kasheel/Zach | *Vitis vinifera* L./RW783 | Constipation (37) & intestinal worms (3) | 2 | 40 | 98 | 0.59 | 0.41 |
| 52 | Gulab | *Rosa indica* L./RW784 | Stomachic (19) & constipation (12) | 2 | 31 | 78 | 0.47 | 0.40 |
| 53 | Kuna | *Chenopodium album* L./RW785 | Constipation (12) & abdominal pain (5) | 2 | 17 | 44 | 0.27 | 0.39 |
| 54 | Aseel Khukunay | *Cicer microphyllum* Benth./RW786 | Indigestion (5) & vomiting (9) | 2 | 14 | 39 | 0.23 | 0.36 |
| 55 | Loi Margan | *Capparis spinosa* L./RW787 | Piles (7) & digestive (5) | 2 | 12 | 35 | 0.21 | 0.34 |
| 56 | Kino Maroch | *Morus nigra* L./RW788 | Stomachic (21) | 1 | 21 | 63 | 0.38 | 0.33 |
| 57 | Hazar Daru | *Limonium cabulicum* (Boiss.) Kuntze/RW789 | Stomachic (21) | 1 | 21 | 75 | 0.45 | 0.28 |
| 58 | Dadi Pushi Char | *Xanthium strumarium* L./RW790 | Ulcer (11) | 1 | 11 | 43 | 0.26 | 0.26 |
| 59 | Pharphara | *Verbascum thapsus* L./RW791 | diarrhea (11) | 1 | 11 | 54 | 0.33 | 0.20 |
| 60 | Ishpit | *Medicago sativa* L./RW792 | Stomachic (12) | 1 | 12 | 65 | 0.39 | 0.18 |
| 61 | Khakos | *Artemisia annua* L./RW793 | diarrhea (3) & vomiting (4) | 2 | 7 | 58 | 0.35 | 0.12 |

Consensus Factor (ICF) were applied for the graphical representation of the numerical data extracted.

**Relative frequency of citation.** The RFC was determined following [36] which expresses the local importance of the plant species. RFC was calculated as a measure of citation frequency that explains the intensity of uses reported for a given species by all the informants It was calculated by dividing the frequency of citation (FC) by the total number of informants involved in the survey (N) as shown below:

$$RFC = FC/N$$

Where FC is the frequency of citation which denotes the number of informants interviewed for a species who site its use while N is the total number of informants involved in the survey.

**Use value.** Use value (UV) expresses the relative importance of the medicinal plant species based on number of recorded uses for each species, known locally. UV is considered a baseline quantitative index in ethnobotanical studies. It was calculated by the methodology given by [37], using following formula:

$$UV = \sum Ui/n$$

Where UV stands for the use value; U refers to the total number of uses for each species; and "n" is the total number of informants who reported that species.

**Fidelity level (FL).** Fidelity level explains the specific use of each plant species and its preference over other species by expressing the specificity of disease treated by a reported plant species. It is applied to distinguish the crucial role of reported species. It is calculated by a formula [38] which is as follows:

$$FL = (Ip/Iu) * 100$$

Where "FL" is the percent fidelity level; "Ip" is the number of informants who mentioned the plant species for the treatment of a particular disease; and "Iu" is the number of informants who mentioned the same plant for any other use.

**Informant consensus factor (ICF).** Informant consensus factor was calculated by following the formula given by [39]. It expresses the consensus of the informants about the use of plant species for the treatment of various diseases.

$$ICF = Nur - Nt/Nur - 1$$

Where "Nur" is the number of use reports for a disease treated by a plant species while "Nt" is the number of plant taxa used to treat that disease. ICF varies from 0–1. It is used to estimate the significance of each medicinal use category centered on the uniformity of the interviewer's response.

**Direct matrix ranking (DMRT).** Direct matrix ranking test (DMRT) was calculated following the method of [32]. It was conducted to compare the species mentioned for multiple uses by the informants. Based on the relative benefits obtained from each plant, ten multipurpose plant species were selected, and seven use diversities of these plants were listed. Five key informants were chosen to assign use values for each attribute (4 = best, 3 = very good, 2 = good, 1 = less, 0 = none). The use categories included medicinal use, fuel, construction, edible, agriculture tools, thatching, fodder, and forage. Based on data obtained from the informants, the average use diversity value for each species was determined and the values for each species were finally summed and ranked.

**Preference ranking (PR).** Ten informants were randomly selected to assess the degree of effectiveness of medicinal plant species when used to treat constipation, a gastrointestinal disease category following the methodology given by [40]. The medicinal plants believed to be most effective to treat constipation were given the highest value (5), while the least effective received the lowest value (1). The value of each species was summed and the rank for each species was determined based on the total score. This helped to indicate the most effective medicinal plants used by the informants to treat a serious gastrointestinal disease (constipation).

*Novelty assessment.* Novelty assessment was the prime objective of present study because it was very likely that the study area owing to its remoteness and peculiar tribal composition would be harboring unique plant use pracitces. From the selected plant species, a few novel plant species were identified with novel use reports from the study area. These species were subjected with a robust comparative review using literature databases viz; google scholar, PubMed and Scifinder to explore any previously reported pharmacological and ethnomedicinal uses of these plants and to compare with the use reports mentioned by locals in present study.

## Results

### Sociodemographic characteristics of the respondents

One hundred and sixty-six (166) informants were interviewed during the ethnomedicinal inventory. Of which, 151 comprising up of 69% were males followed by 51 contributing 31% females. Of these, 40(24%) informants were selected from Thamros village, 36(22%) from Gashot, 30(18%) from Pakora, 27(16%) from Buner Das, 18(11%) from Halaly, and 15(9%) from Keli Jail. Age wise, 10% of the informants had 20 to 30 years of age, 14% had 31–40 years, 24% had 41–50 years, 27 had 51–60 years and 26% of the informants had above 60 years of age. It was observed that age of the informants greatly affects the ethnomedicinal information as the older people have more experience and knowledge in this regard. The marital status of the informants was also noted. It was observed that 123(74%) of the informants were married and 36(22%) were unmarried while 7(4%) were widowed.

Majority of the informants were illiterate 97(58%) followed by elementary school 25(15%), secondary school 22(13%), college 15(9%) and university students 7(4%). Due to remoteness of the study area, majority of the people rely on agriculture and livestock practices for their livelihood. Out of 166, 112(68%) of the informants were farmers which is followed by shepherd 31(19%), retired persons 5(3%) and others 18(11%). Most of the people 155(93%) were residing in villages and only 11(7%) were seasonal migrants who live both in villages and cities in different seasons of the year. Four different ethnic groups were residing in the study area among which 79(48%) were youshkon, 34(20%) sheen, 38(23%) gujar and 15(9%) were Kashmiri. From the study area, 7(4%) of the informants have experience in herbal medications and they were practicing in cities as herbalists and 159(96%) have no proper experience in herbalism and they self-medicate of their illness with medicinal plants. A very few 9(5%) informants were residing in the study area from less than 15 year while 157(95%) were living there from more than 15 years. Religiously, 100% of the informants were Muslim (**Tab.1**)

### Diversity of medicinal plants for treatment of gastrointestinal disorders

A total of 61 medicinal plant species belonging to 55 genera and 35 families have been documented from the study area for the treatment of GIDs. Compositae was the most dominant family with 8(13%) species followed by Polygonaceae 6(10%), Lamiaceae, Leguminosae, Moraceae and Rosaceae 3(5%) each, Apiaceae, Cucurbitaceae, Poaceae and Solanaceae 2(3%) each and all other species were contributing 1(2%) (**Fig 2A**. Fourteen different gastrointestinal disorders have been treated by the collected medicinal plants of which stomach disorder was the leading as 33(54.1%) plant species were used for its treatment followed by diarrhea 16(26.2%), constipation 14(23%), intestinal worms 9(14.8%), vomiting 7(11.5%), digestive and ulcer 6(9.8%) each, indigestion and abdominal pain 3(4.9%) each, gastritis and piles 2(3.3%) each while dysentery, purgative and flatulence each were treated with 1(1.6%) of the collected medicinal plants (**Fig 2B**).

### Phytogeographic distribution and utilization status of medicinal plants

The plant collection was carried out in a wide range of altitude ranging from subtropical limatic conditions adjacent to the river Indus to the timber line in upper alpine mountains. Some of the herbaceous plants such as *V. serpens* and *H. telephioides* were collected above the timber line. The study areas were designated via an improvised classification into plains (mostly narrow terraces along the river or tributary streams), foothills such as (The Deosai plain) and the mountains (S1 Table). Moreover, the availability of plant species was also

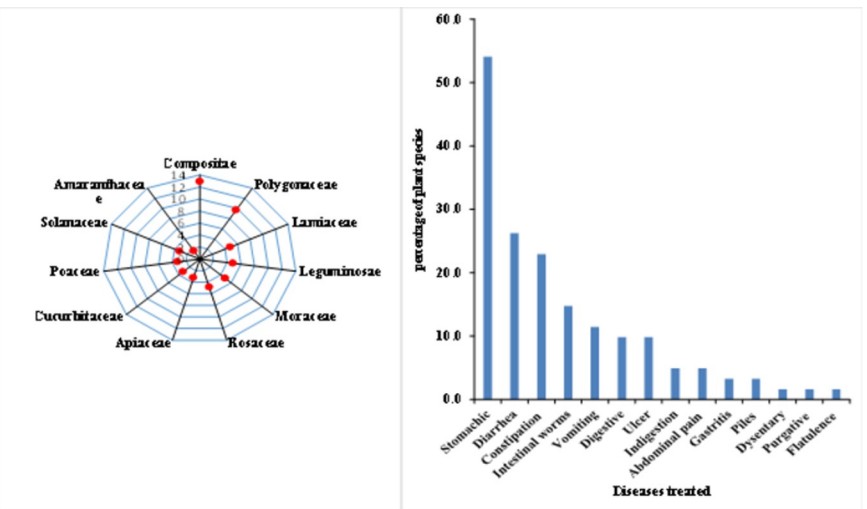

**Fig 2.** (a) Dominant families in the study area given as a measure of floral diversity, (b) Various gastrointestinal disorders treated with medicinal plants in the study area.

recorded as a plant species was flagged as available frequently, occasionally, and rarely. This somehow indicates the conservation status of the plant species from an informant eye.

## Methods of preparation

The collected medicinal plants were locally used by various preparation methods. Majority of the plant and plant parts were used in the form of powder contributing 27(34%) followed by direct 19(24%), juice 10(13%), paste 8(10%), roasted form 7(9%), decoction 4(5%), infusion 3 (4%) and smoke 1(1%) with least preparation method (**Fig 3A**). People in the study area directly collect the plant parts and dry either under shade or sun and make powder that is utilized with either water, milk, honey or mixed with any other edible material. The excessive use of powdered material was because of its easy use and storage for longer periods.

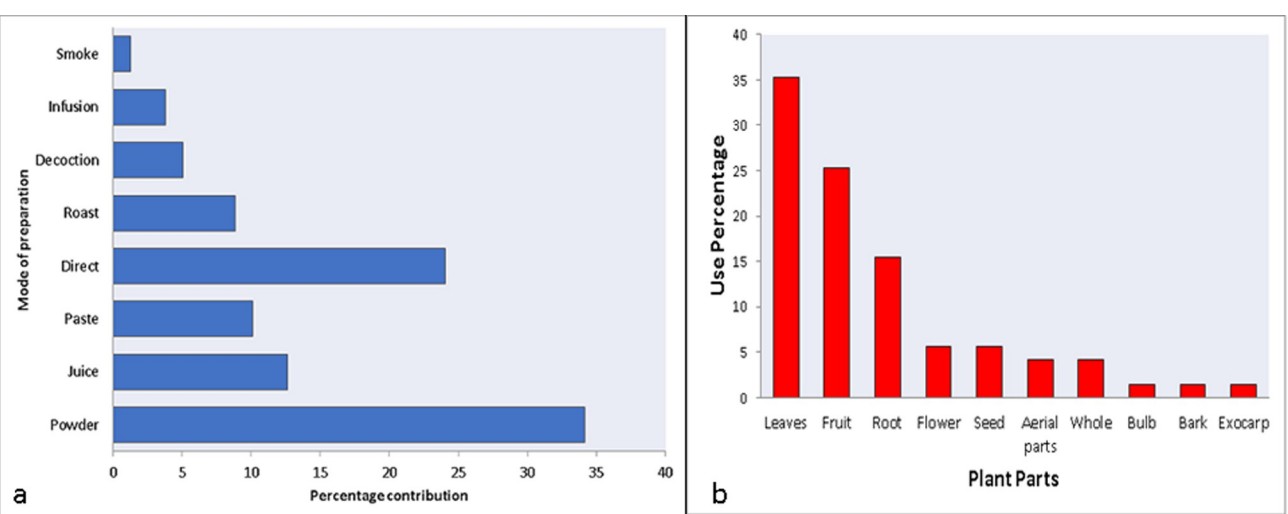

**Fig 3.** (a) Methods of Preparation of folk recipes, (b) Different plant parts used for the treatment of gastrointestinal disorders.

## Plant part(s) used for the treatment of gastrointestinal disorders

Different plant part(s) have been used by the locals to treat their GIDs in the study area. Among all, leaves were the dominant parts contributing 25(35%) followed by fruit 18(25), root 11(16%), flower and seed each 4(6%), aerial parts and whole plant 3(4%) and remaining were contributing 1(1%) shown (**Fig 3B**). Leaves were mostly collected and used by the locals because it is easy to collect the leaves, and these are the most abundantly available parts of the plant. In addition to these leaves are the major photosynthetic machinery which makes almost all the phytochemical in it and due to this, leaves are the most effective plant parts [41].

All the collected plant parts were used orally. Based on the field visits and collection of the specimens, occurrence of the plant species was also mentioned. It was observed that 23(37%) of the medicinal plants were found in mountains followed by 19(31%) in both foothills and plains, 11(18%) species in plains, 4(7%) species in foothills and mountains, 3(5%) were common in all plains, foothills, and mountains while 2(3%) of the plants were restricted to the foothills only (S1 Table). The collected plant species were used in various use forms as 30(48%) of the plant species were used in fresh form followed by dried form 19(31%) and 13(21%) both in fresh and dried form. The collected medicinal plant species were grouped into various life forms as 50(80.6%) were herbs followed by trees 8(12.93%), shrubs 3(4.8%) and climbers 1 (1.6%) (S1 Table).

From the collected medicinal plants, ten plants were selected for the preference ranking from ten randomly selected informants. It was observed that *H. telephioides* (Ledeb.) was the most effective plant to treat GIDs (constipation) and it was assigned with a value 5 by all the key informants and it stood at rank first. Similarly, *A. maritima* L. was ranked second, *Pimpinella diversifolia* DC. ranked third and all other plants below this rank (S1 Table). The output of the preference ranking indicated that *H. telephioides*, *A. maritima* and *P. diversifolia* were the most preferable medicinal plants for the treatment of constipation. Direct matrix ranking test was also used to evaluate the functionality of multipurpose uses of the ten randomly selected medicinal plant species mentioned by the five key informants against eight usage categories. The key informants assigned the use values/scores for each plant species (on a scale of 1–5) and categorize each plant. We observed that *Prunus persica* L. ranked first followed by *Ficus carica* L., *Prunus armeniaca* L. each ranked second, and all other plant species ranks were shown in (**Fig 4**).

The percentage contribution of the use citation of fifteen medicinal plant species representing 50% of use reports/citations was also measured. For this purpose, the use citations of each plant species for various GIDs were summed and its percentage was calculated. It was observed that *M. longifolia* contributed the highest use citations with 6.4% for overall use citations followed by *A. maritima* 6.2%, *Tanacetum falconeri* 5.8%, *H. telephioides* 4.1%, *M. piperita* 3.4% while all other plant species contributed below this rang (**Fig 5**).

## Fidelity level percentage

Fidelity level for fourteen different GIDs was measured against the collected plant species. It was observed that fidelity level percentage was varied from 100–10 for different plant species. A maximum of 100% fidelity was obtained for six plant species including *A. maritima*, *M. longifolia*, *M. piperita*, *A. annua*, *H. telephioides* and *Allium cepa* against diarrhea and constipation while least fidelity level of 9.8% was shown by *Chenopodium album* against dysentery (**S2 Table**). The highest fidelity level percentage denotes that the plant species is highly preferred for that disease while lower fidelity level indicates that the plant species is not preferred by the informants.

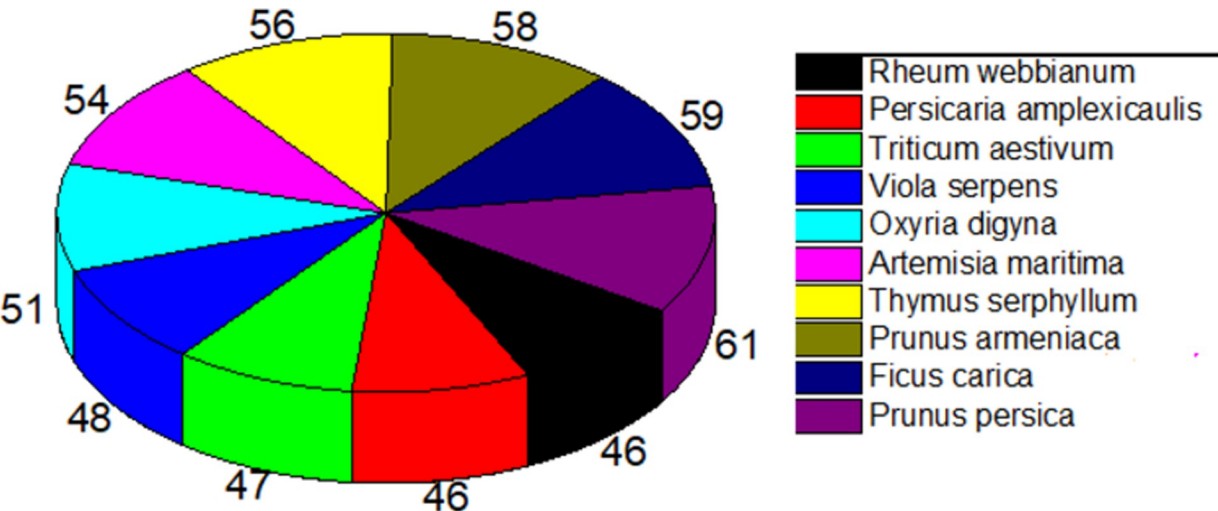

**Fig 4. Direct Matrix Ranking (DMRT) of selected plant species to evaluate the functionality of multipurpose uses of randomly selected medicinal plant species mentioned by key informants against eight usage categories.**

## Informant consensus factor

Present study was focused on the medicinal plants used for the treatment of various GIDs. Fourteen different gastrointestinal disorders were cured from 61 collected medicinal plant species by the local community. Among these diseases, the categories with highest ICF value were dysentery, purgative, and flatulence (1.0) each followed by piles, diarrhea, and vomiting (0.98), intestinal worms, constipation, gastritis, stomachic, indigestion and digestive (0.97) while least ICF was calculated for abdominal pain (0.94) (Table 2). The ICF shows that the plant species with highest ICF values were presumed to be more effective and common when used to treat certain disease. Lower ICF values indicated that the informants disagreed on the taxa to be used as a treatment within the disease category. The total use citations for all the plant species

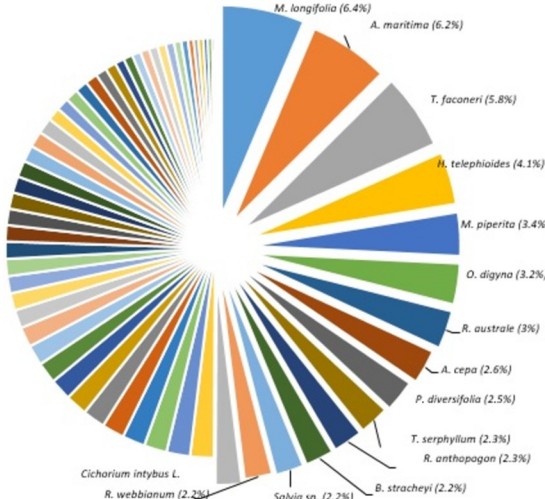

**Fig 5. List of 15 most cited species representing 50% of use reports/use citations.** Rest of the pie-chart represents 46 plant species.

that have been used to cure various GIDs was 3248 with highest use citation for stomachic and stomach problems (1150) followed by diarrhea (624) while least use citations were found against flatulence (22) and dysentery (9) (Table 2).

## Novelty assessment

The ethnomedicinal data about the GIDs obtained from the study area was compared with the previously published data in Pakistan and neighboring countries to explore the novelty of the plant-based use data of the region. The comparison of the data was broadened with other countries for a much-explored novelty assessment. The comparative analysis identified 19 plant species with specific use reports against different gastrointestinal disorders from the study area for the first time. Among these, *T. falconeri*, *A. maritima* & *M. piperita* were reported to cure vomiting, *T. falconeri*, *R. australe*, *O. digyna and Viola serpens* were identified against stomachic again *T. falconeri* and *V. serpens* were reported against diarrhea while *H. telephioides*, *R. australe* and *O. digyna* were identified with novel use to cure constipation. The pharmacological review of plant species was developed with an extensive literature review (Table 3).

## Discussion

From the study area, 61 plant species were identified to be used for the GIDs. Among these, maximum number of plants were being used for stomachache followed by diarrhea, constipation, and intestinal worms. Previous studies also confirmed the treatment of GIDs using local medicinal plants [12, 67–70]. The results suggest that locals prefer plant-based remedies for the treatment of GIDs, as in most of the cases herbal preparations provide instant relief without any significant side effect. One of the associated reasons for excessive use of plants for the treatment of GIDs is taboos linked to certain GIDs. For example, people often hesitate to share problems of bad breadth, intestinal gas, and piles comfortably to physicians. Instead, they consult family elders who recommend them to use a locally available medicinal plants or an herbal preparation made up of multiple plants as per their folklore. Other reasons include remoteness of the area and lack of modern health care facilities. It was observed during the survey that there is only one district headquarter hospital (DHQ) situated in Chilas city that is about 23 km away from Buner Das and 38–43 km away from the rest of the study area. In addition, there is no direct road access to the DHQ from the study area instead people travel on foot several miles to get access to the roads. However, there are three dispensaries also exist in the study area that hardly provide primary health care medications for a few commonly occurring ailments.

We calculated FL of each species to quantify relative trust of locals on a particular plant for treating a specific disease. A maximum of 100% fidelity was obtained for six plant species including *H. telephioides*, *A. maritima*, *M. longifolia*, *M. piperita*, *A. annua* and *Allium cepa* against constipation and diarrhea. The plant species having highest FL values indicates a good healing potential against the mentioned disease [71–75]. Based on these findings, we attempted to identify the most preferred plant species with trusted use practices prevailing for treatment of constipation and diarrhea in the study area. *H. telephioides* for example ranked first for all ten informants selected randomly for preference ranking for the treatment of constipation. Informants consensus factor was also established which indicates the consensus among informants for the treatment of diseases [72]. Among the diseases, highest ICF value was reported for dysentery and flatulence followed by piles, diarrhea, and vomiting and constipation. Higher ICF values indicate consensus in use reports among informants for the treatment of a

**Table 3. Novelty assessment of the medicinal plants with novel uses against specific gastrointestinal disorders(bold).**

| S. No. | Plant name | Local use | Previously reported uses | Reference |
|---|---|---|---|---|
| 1 | *Artemisia maritima* L./RW735 | Intestinal worms, diarrhea, **vomiting** & stomachic | Antiseptic, anthelmintic, antidiabetic, antihypertensive, emmenagogue, antivenom, digestive & cutaneous problems, diarrhea, nausea, fever, Asthma, headache, stomachic, anti-inflammatory, antimalarial, cooling purposes, intestinal worms | [42, 43] |
| 2 | *Tanacetum falconeri* Hook.f./RW739 | **Diarrhea, vomiting & stomachic** | Abdominal problems, asthma, blood pressure & jaundice | [44] |
| 3 | *Mentha piperita* L./RW736 | Diarrhea & **vomiting** | Antidiabetic, antioxidant, hepatoprotective, antipyretic, antispasmodic, analgesic, carminative, diaphoretic, analgesic, anti-diarrhea, anti-microbial, treatment of irritable bowel syndrome (IBS), inflammatory bowel disease, inflammation & dysfunction of the gallbladder & liver diseases | [45–48] |
| 4 | *Hylotelephium telephioides* (Ledeb.) H. Ohba/RW741 | **Constipation** | Skin disease, earache & wounds | [49] |
| 5 | *Oxyria digyna* (L.) Hill/RW740 | **diarrhea, constipation & stomachic** | Appetite disorder, improves digestion | [41, 50] |
| 6 | *Viola serpens* WalL./RW748 | **diarrhea**, constipation & **stomachic** | Antioxidant, antipyretic, demulcent, diaphoretic, diuretic, asthma, bleeding piles, cancer of throat, cough, fever, skin diseases, headache, emollient, expectorant, febrifuge, purgative, constipation, cough, fever & antinociceptive | [51, 52] |
| 7 | *Persicaria amplexicaulis* (D. Don) Ronse Decr./RW749 | **Stomach disorders** | Sores, wounds, to check excess bleeding during menstruation period, dysentery, cough, rhematic pain, backache, gout, eyesight, purify blood & cause abortion | [53] |
| 8 | *Saussurea gossypiphora* D. Don/RW742 | **constipation** | Asthma, pneumonia, stomach problem, flue, headache, improve circulation, antioxidant, anti-inflammatory and antibacterial, evil spirits, menstrual disorders, hysteria & wound | [54] |
| 9 | *Aconitum heterophyllum* Wall. ex Royle/RW759 | Stomachic & **intestinal worms** | Reduce bile, headache, cold, fever, stomachic, urinary infections, diarrhea, inflammation, antidiarrheal, expectorant, diuretic, hepatoprotective, antipyretic, analgesic, antioxidant, alexipharmic, anodyne, anti-atrabilious, anti-flatulent, anti-periodic, anti-phlegmatic, carminative properties, reproductive disorders, diarrhea, liver, spleen, urinary tract diseases & diabetic | [55] |
| 10 | *Ribes alpestre* Wall.ex Decne./RW764 | **Ulcer** | Antiarthritic, backache, joints pain, jaundice, liver problems, fever, burns, blisters & coolant | [23] |
| 11 | *Echinops echinatus* Roxb./RW770 | **Abdominal pain** | Sexual debility, antimicrobial, analgesic, diuretic, reproductive, hepatoprotective, antioxidant, anti-inflammatory, wound-healing, antipyretic, spermatorrhea, worms, Aphrodisiac, abortifacient, leukorrhea, diabetes, diarrhea, jaundice, hysteria, dyspepsia, hoarseness of throat, cough, asthma, chronic fever, migraine, heart diseases, joint pains, cardiac diseases, lice, ticks, teeth infection, stomachic, analgesic, urinary disorder & hemorrhoids | [56] |
| 12 | *Rumex hastatus* D. Don/RW771 | **Stomachic & flatulence** | Headache, migraine, depression, paralysis, antioxidant, anticholinesterases potentials, laxative, alterative, tonic, rheumatism, skin diseases, piles, bilious complaints, lungs bleeding, blood pressure, tonsillitis, sore throat, anti-poison, digestion, antirusting. flavoring, carminative, diuretic, giddiness & insanity | [57] |
| 13 | *Pistacia khinjuk* Stocks/RW773 | **Indigestion** | Tonic, aphrodisiac, antiseptic, antihypertensive, gastrointestinal, liver, urinary tract, respiratory tract disorders, antioxidant, antimicrobial, antiviral, anticholinesterase, anti-inflammatory, antinociceptive, antidiabetic, antitumor, antihyperlipidemic, antiatherosclerosis, hepatoprotective | [58] |
| 14 | *Plantago himalaica* Pilg./RW779 | **Stomachic & diarrhea** | Dysentery & Stops bleeding from nose | [41, 59] |
| 15 | *Vitis vinifera* L./RW783 | Constipation & **intestinal worms** | Laxative, purgative, diuretic, aphrodisiac, fever, asthma, jaundice, vomiting, stomach problems, piles, joint aches, expectorant, liver tonic, bronchitis, constipation, watering of eyes, lung cancer, burn, boil, wound care, anemia, bronchitis, cold, flu, carminative, costiveness, intestinal spasm, dyspepsia & allergy | [60] |
| 16 | *Cicer microphyllum* Benth./RW786 | **Indigestion & vomiting** | Abscess, swelling of the limbs, poisoning, spleen disorder, colic pain, tonic, mouth & khur disease in goat/sheep | [61] |
| 17 | *Capparis spinosa* L./RW787 | **Piles & digestive** | Toothache, fever, headache, menstruation, rheumatism, convulsion, gout, skin disease, kidney, liver, diabetes, hemorrhoids, ulcers, sciatica, chest disease, febrifuge, dropsy, colds, backache, feminine sterility & dysmenorrheal, anti-dandruff & arthritis | [62, 63] |

*(Continued)*

**Table 3.** (Continued)

| S. No. | Plant name | Local use | Previously reported uses | Reference |
|---|---|---|---|---|
| *18* | *Artemisia annua* L./RW793 | Diarrhea & **vomiting** | Antimalarial, anti-inflammatory, anti-microbial, fever, chills, wound healing, intermittent fevers, jaundice, sedative, diarrhea, Anemia, Asthma, eye infections, Cholera, dengue fever, athlete's foot & eczema, chagas disease, viral hepatitis B & schistosomiasis | [64] |
| *19* | *Rheum australe* D. Don/RW743 | **stomachic & constipation** | Renal function disorders, hyperlipidemia, cancer, digestive problem, diarrhea, laxative, wounds, headache, body pain, joint pain, cure boils, appetizer, astringent, purgative, and health tonic | [65, 66] |

particular disease. These results suggest that lower intestinal disorders were more commonly prevailed and subsequently the local populace had indigenous healing options in practice.

Owing to poorly explored nature of study area, we hypothesized that these valleys might be hosting some novel medicinal uses of plants. So, we compared our finding systematically with previously published studies in the region at large. It was a pleasant surprise that to best of comparative analysis we carried out, we found 19 medicinal plant species with novel use reports against different GIDs. In our study, for example, fresh leaf juice of *A. maritima* mixed with mint leaves is taken orally to cure vomiting. The pharmacological review of this plant showed that it has been used as antiseptic, anthelmintic, antidiabetic, antihypertensive, emmenagogue, antivenom, digestive & cutaneous problems, diarrhea, nausea, fever, Asthma, headache, stomachic, anti-inflammatory, antimalarial, cooling purposes and intestinal worms [42, 43].

Fresh leaf juice of *T. falconeri* grinded with mint leaves is taken orally for the treatment of diarrhea, vomiting and stomachic. The same plant has been reported to be used against abdominal problems, asthma, blood pressure and jaundice [44]. *Hylotelephium telephioides* is another important plant used to cure against constipation. Half a teaspoon or 5–10 g of the fresh plant part (s) mixed either with water, milk or cured and taken orally. Literature revealed that the plant has been used to cure skin disease, earache, and wounds [49]. Fresh and dried leaves of *M. piperita* are used as paste, juice and powder mixed with *Allium cepa* and *A. maritima* leaves for the treatment of vomiting and also reported previously as antidiabetic, antioxidant, hepatoprotective, antipyretic, antispasmodic, analgesic, carminative, diaphoretic, analgesic, anti-diarrhea, anti-microbial, treatment of irritable bowel syndrome (IBS), inflammatory bowel disease, inflammation and dysfunction of the gall bladder and for liver diseases [45–48]. Dried root powder of *R. australe*, mixed with butter or milk is taken orally once a day for to cure stomachic and constipation. The same plant was reported from different parts of the world for the treatment of renal function disorders, hyperlipidemia, cancer, digestive problem, diarrhea, laxative, wounds, headache, body pain, joint pain, cure boils, appetizer, astringent, purgative, and health tonic [65, 66]. Fresh leaves of *Oxyria digyna*, is another important medicinal plant. used to cure diarrhea, constipation and stomachic. the same plant has been reported to be used against appetite disorder and to improves digestion [41, 50]. Present study reported that fresh leaves and fruit of *Viola serpens* for diarrhea and stomachic. The plant is previously reported as antioxidant, antipyretic, demulcent, diaphoretic, diuretic, cough, asthma, piles, throat cancer, constipation, skin diseases, headache, as emollient, expectorant, febrifuge, purgative, and antinociceptive [51, 52]. Dried root powder of *Persicaria amplexicaulis* taken orally with butter for stomach disorders in the study area the plant has been reported from the allied valley for the treatment of sores, wounds, dysentery, cough, rhematic pain, backache, gout and to control excess bleeding during menstruation period [53].

The decoction of *Saussurea gossypiphora* (dried flower) is taken daily before bed and early in the morning to cure constipation. The plant is previously reported to cure asthma, pneumonia, stomach problem, flu, headache, and is used to improve blood circulation, as antioxidant, anti-inflammatory and antibacterial, menstrual disorders, hysteria and wound [54]. *Aconitum heterophyllum is* used for the treatment of intestinal worms. The plant is reported to treat reproductive disorders, diarrhea, liver, spleen, urinary tract diseases and diabetic, headache, cold, fever, stomachic, urinary infections, diarrhea, inflammation, and as expectorant, diuretic, hepatoprotective, antipyretic, analgesic, antioxidant, alexipharmic, anodyne, anti-atrabilious, anti-flatulent, anti-periodic, anti-phlegmatic, carminative [55].

Another important plant species fresh fruit of *Ribes alpestre* is taken orally to cure stomach ulcer. The plant is already reported from allied areas as antiarthritic, coolant and to treat backache, joints pain, jaundice, liver problems, fever, burns, blisters and [23]. Fresh leaf infusion of *Echinopse chinatus* has been used by the locals against abdominal pain. Same plant was reported in other parts of the world against sexual debility and as antimicrobial, analgesic, diuretic, reproductive, hepatoprotective, antioxidant, anti-inflammatory, wound-healing agent, antipyretic, aphrodisiac, abortifacient analgesic, leukorrhea, diabetes, diarrhea, jaundice, hysteria, dyspepsia, hoarseness of throat, cough, asthma, chronic fever, migraine, heart diseases, joint pains, cardiac diseases, teeth infection, stomachic, urinary disorder and hemorrhoids [41, 56]. Fresh aerial parts of *Rumex hastatus* are collected and grinded with water to obtain juice which is taken orally to cure flatulence and as stomachic. Literature have revealed that, above plant has been used against headache, migraine, depression, paralysis, antioxidant, anticholinesterases potentials, laxative, alterative, tonic, rheumatism, skin diseases, piles, bilious complaints, lungs bleeding, blood pressure, tonsillitis, sore throat, anti-poison, digestion, antirusting. flavoring, carminative, diuretic, giddiness and insanity [57].

Gum of *Pistacia khinjuk* mixed with honey is taken orally for indigestion. Literature reported that the plant is used as tonic, aphrodisiac, antiseptic, antihypertensive and for GIDs, urinary and respiratory tract infections, as antioxidant, antimicrobial, antiviral, anticholinesterase, anti-inflammatory, antinociceptive, antidiabetic, antitumor, antihyperlipidemic, anti-atherosclerosis and hepatoprotective [58].

Local people in the study area use juice and paste of *Plantago himalaica* leaves against stomachic and diarrhea. The plant has also been reported in literature to be used against dysentery and stops nose bleeding [41, 59]. Fresh leaves and fruit of *Vitis vinifera* taken orally by locals to cure intestinal worms where, literature review has revealed that the plant has been used as laxative, purgative, diuretic, aphrodisiac, fever, asthma, jaundice, vomiting, stomach problems, piles, joint aches, expectorant, liver tonic, bronchitis, constipation, watering of eyes, lung cancer, burn, boil, wound care, anemia, bronchitis, cold, flu, carminative, costiveness, intestinal spasm, dyspepsia and allergy [60]. Fresh leaves of *Cicer microphyllum* is used against vomiting and indigestion in the study area while literature have revealed that it is used against abscess, swelling of the limbs, poisoning, spleen disorder, colic pain, tonic, mouth & khur disease in goat/sheep [61]. Shade dried seed powder of *Capparis spinosa* is mixed with milk or water and taken orally once a week to cure piles and as digestive. Literature reported that the same plant has been used to cure toothache, fever, headache, menstrual cycle disturbance, rheumatism, convulsions, gout, and skin disease. The plant is also reported for kidney problems, liver related ailments, diabetes, hemorrhoids, ulcers, sciatica, dropsy, colds, backache and to cure feminine sterility [62, 63]. Paste of the fresh leaves of *Artemisia annua* is used orally by the locals to cure serious vomiting while it has been reported in literature for as anti-malarial, anti-inflammatory, anti-microbial wound healer and to cure fever, chills, intermittent fevers, jaundice, diarrhea, Anemia, Asthma, eye infections, Cholera, dengue fever, athlete's foot, eczema, chagas disease, viral hepatitis, schistosomiasis and as sedative [64].

Recently, GIDs have got the attention of scientific community and a new conception of brain-gut-functional axis has emerged to maintain nature's wellbeing equilibrium [76]. The role of plants and their derivatives is somehow reiterated not only for the treatment of GIDs but also for mitigation of various neurological disorders and psychological condition known to be aggravated by gut discomfort [3, 5, 76, 77]. It is therefore, a need of the hour to extend investigation of aforementioned plants to deduce tangible results for human wellbeing in a holistic manner.

## Conclusion

For the first time, this study provided the information on 61 species of traditional applications for the treatment of GIDs by the inhabitants of Diamir, Gilgit, Pakistan. It further enriches our knowledge about the medicinal plant potential and traditional medicinal knowledge of locals about treatment options for GIDs in one of the most far-flung yet iconic region known for its adventurous trekking and mountain climbing. Assessment of conservation status of local flora particularly the plant species with medicinal novelties is highly recommended before any mass collection drive. However, pharmacological evaluation of medicinal flora, with novel therapeutic uses is merited on experimental scale for potential drug leads. Assessment of conservation status of local flora particularly the plant species with medicinal novelties is highly recommended before any mass collection drive. However, pharmacological evaluation of medicinal flora, with novel therapeutic uses is merited on experimental scale for potential drug leads.

## Supporting information

**S1 Table. Supplementry data file 1 (inventory, basic data sheet).**
(PDF)

**S2 Table. Supplementry data file 2 (fidelity level).**
(PDF)

## Acknowledgments

Locals of the study area are acknowledged for TMK and for their stewardship in conservation of centuries old medicinal practices community stakeholders.

## Author Contributions

**Conceptualization:** Zia-ur-Rehman Mashwani.

**Data curation:** Muhammad Faraz Khan.

**Formal analysis:** Muhammad Faraz Khan.

**Investigation:** Rahmat Wali.

**Software:** Ansar Mahmood.

**Supervision:** Rahmatullah Qureshi, Zia-ur-Rehman Mashwani.

**Validation:** Rahmatullah Qureshi.

**Visualization:** Majid Mahmood.

**Writing – original draft:** Rahmat Wali, Muhammad Faraz Khan.

**Writing – review & editing:** Ansar Mahmood, Majid Mahmood, Khawaja Shafique Ahmad.

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
