## [Decision Letter · Decision Letter 0]

28 Mar 2022

PONE-D-22-02797Ethnomedicinal appraisal of Plants used for the treatment of gastrointestinal complaints by tribal communities living in Diamir district, Western Himalayas, PakistanPLOS ONE

Dear Dr. Khan,

Thank you for submitting your manuscript to PLOS ONE. After careful consideration, we feel that it has merit but does not fully meet PLOS ONE’s publication criteria as it currently stands. Therefore, we invite you to submit a revised version of the manuscript that addresses the points raised during the review process.

We look forward to receiving your revised manuscript.

Kind regards,

Jen-Tsung Chen, Ph.D.

Academic Editor

PLOS ONE

Journal Requirements:

Reviewers' comments:

Reviewer's Responses to Questions

**Comments to the Author**

1. Is the manuscript technically sound, and do the data support the conclusions?

Reviewer #1: Yes

Reviewer #2: Yes

Reviewer #3: Yes

2. Has the statistical analysis been performed appropriately and rigorously? 

Reviewer #1: N/A

Reviewer #2: Yes

Reviewer #3: I Don't Know

3. Have the authors made all data underlying the findings in their manuscript fully available?

Reviewer #1: Yes

Reviewer #2: Yes

Reviewer #3: Yes

4. Is the manuscript presented in an intelligible fashion and written in standard English?

Reviewer #1: Yes

Reviewer #2: Yes

Reviewer #3: Yes

5. Review Comments to the Author

Reviewer #1: In the present study, the authors have done extensive work in conducting field surveys, and collecting plants and information on traditional use of these plants against gastrointestinal disorders in a remote area of not only Pakistan but possibly also the entire world. The remoteness makes the survey interesting from two points. The first is that new plant species or new uses of already known plant species can be discovered, which has precisely happened in this case. The second is that the remoteness of the area is an impediment to introduction of allopathic medicine, modern hospitals and diagnostic centers; this would suggest that the plants have a possibility of long history of usage and so the medical uses of any plant increase in reliability through their long history of use. Any toxicity or other adverse effects of the plant would have been noted in use over a prolonged time period. On the other hand, the toxicity problem cannot totally be overlooked. Datura stramonium was reported by the authors to be used as a stomachic. The plant, although having analgesic properties, is also very toxic and so exact dose needs to be determined by an experienced practitioner.

Despite the novelties described in the manuscript, several important things are missing, which could have added much necessary information. For instance in Table 3, to assess novelty, nineteen plants have been assessed comparing their use in the survey area versus other reported uses. However, there is no information on the plant part(s) used in the survey area versus other reported areas. Different plant parts (shoots, roots, flowers, seeds) may contain different phytochemicals with different pharmacological properties and medicinal uses. So while the use report may appear to be different in various areas, this may be a consequence of different parts of the plant being used in differing areas.

No detailed information has been given on how the plants were collected; whether they were used in fresh or dried state; if dry then how long can the medicinal activities of a given plant be sustained; preparation of medicine like powder, decoction or other forms, dosage, part of plant used, use of monoherbal or polyherbal formulation; if the same plant is used for more than one gastrointestinal disorder like say diarrhea and constipation did the mode of use differed and how; endangerment of these medicinal plants and any efforts by the locals in their conservation; and the altitude range where the plants can be found.

Gastrointestinal disorders are common in rural areas of practically every developing country. As such, thorough studies of the plants used traditionally to treat such disorders are important, more so because people residing in these areas lack access to modern medical facilities. The authors note that elderly people possessed more knowledge of traditional plant usage. In that case, why were more elderly people not consulted to gather more information? That elderly people know more of their traditional customs is open knowledge; the question is how can the young generation be made more interested in the traditional customs? A short discussion on this topic can also be a valuable addition to the manuscript. An improved version of this manuscript incorporating the changes described above can be of considerable importance to the ethnomedicinal literature.

Reviewer #2: Review report

Dear editor and authors,

The interesting manuscript titled as “Ethnomedicinal appraisal of Plants used for the treatment of gastrointestinal complaints by tribal communities living in Diamir district, Western Himalayas, Pakistan is a novel and significant contribution to science. In this work, Wali and coworkers have performed an extensive ethnobotanical study on the medicinal plants to treat the gastrointestinal complaints by tribal communities living in Diamir district, Western Himalayas, Pakistan. The manuscript is simple, well-organized and easy to read. The theory is also easy to understand and I think it will be read by many researchers. However, before the paper can be considered for publication/acceptance, it is necessary for authors to undertake minor revisions in accordance with the comments of as suggested by me. I hope that this article will appeal wide readership attention and will accelerate progress in this specific field. However, I do find some minor mistakes/shortcomings in the article that the author need to correct before publication. The manuscript is well written but I would recommend revision and improvement of certain sentences. I found this article insightful, applied and quite informative and recommend it for publication with minor revision.

The abstract is well-structured, well written and has excellently discussed and concluded the different aspects. However, it would be highly appreciated if authors incorporate sentence about the future perspectives of the research work conducted from present study.

Add the word “The” before “resent study was conducted in fairy meadows” in the abstract.

There is no future perspective statement in the abstract section. Provide a statement that what next can be done after these ethnobotanical surveys.

The introduction section is quite informative and to the point. Authors have comprehensively discussed the notion behind present research work. Authors are advised once again to double check grammar, sentence structure etc, if there any deficiency, fix them accordingly.

Provide updated literature to the statement “According to world health organization (WHO) estimates…………. million deaths on 2012 alone. Update 2021-2022 statistics if available.

There is no discussion on the overall medicinal plants flora of the said geographical area. Provide few words and discuss about the dominant medicinal plants.

What was ecological distribution and phytogeography, utilization status and threats to the surveyed medicinal plants? Do you have some recommendations regarding their conservation aspects?. I think you must add 1-2 lines.

Great efforts have been made to write Materials and methods section. All protocols have been explained with full details which is quite fascinating for future researchers to easily pursue their experimental research data collected from field.

Results and discussion section is well written, all results have been rightly discussed with relavent data from literature. Add some new 2020-2021 data from literature if available.

The conclusion section is comprehensive and well-articulated. Remove results and introductory statements from conclusion section after carefully reviewing manuscript.

There is no uniformity in references. Cross check all the references and strictly follow PLOS ONE author guidelines.

Reviewer #3: The manuscript entitled “Ethnomedicinal appraisal of Plants used for the treatment of gastrointestinal complaints by tribal communities living in Diamir district, Western Himalayas, Pakistan”. Authors have analysed and discussed the traditional plant use practices for the treatment of gastrointestinal disorders (GIDs) in Diamir Pakistan Himalayas. They were reported the total of 61 medicinal plant species belonging to 55 genera and 35 families for the treatment of GIDs. Among these, Present study results revealed that 19 out of 61 plant species were documented for the first time with novel medicinal uses to combat GIDs.

The manuscript comprises all the necessary elements of scientific novelty. The study designing and execution of the study were appreciable. I recommend this manuscript for the reconsideration for publication in PlosOne after incorporating minor changes given in below.

COMMENTS FOR THE AUTHOR:

Authors must concentrate on the formatting, and use of symbols, etc., throughout the manuscript.

Continuous line numbers are needed for point out the syntax error, corrections in the manuscript.

Keywords should be unbold.

In introduction, authors have cited the WHO report 2017 it should be referred and cite after 2019 WHO report is better.

Authors should refer few more GIDs articles and include some lines in the introduction section. Because the GIDs information is looks shallow. Better to discuss deeply.

Institutional ethical committee number should be added in the statement of ethics section.

Remove the dot in front of lines in Documentation of Ethnomedicinal Data section.

In sub-title of Results and Discussion section should be “Results”. Because authors have written the separate discussion section.

Framework figure is required. It will be useful to the readers for better understanding of the studied issue.

In materials and method section: authors should explain why each item of methodology was done.

Proper formatting needed in the materials and methods, results, discussion and conclusion section.

In conclusion section authors should provide some future prospectus related to present study.

Reference section should be formatted according to the journal.

Authors should clearly state the figure caption as well as description for ease of understanding of the readers.

6. PLOS authors have the option to publish the peer review history of their article (what does this mean?). If published, this will include your full peer review and any attached files.

Reviewer #1: **Yes: **Mohammed Rahmatullah

Reviewer #2: No

Reviewer #3: No

---

## [Author Response · Author response to Decision Letter 0]

11 May 2022

Response to the reviewer’s comments

 Response to reviewer 1 

S. No Reviewer Comment Response

1 R1 Toxicity or other adverse effects of the plant would have been noted

Exact dose of the Datura stramonium needs to be determined. Our survey was somehow targeted to gather plant use information specially for the treatment of GIDs that’s why information on toxicity or adverse effects was not recorded however most of the non-practicing informants and all of the herbal practitioners stressed on use of botanicals in very quantities and this information is well recorded in method of preparation section for example the dose of Datura stramonium “one to two teaspoons of dried seed powder are mixed with 500 ml of water and taken once a week” (Supplementary file 1; row 44, Column M)

2 R1 Need information regarding the use of plant part(s) in the survey area versus other reported areas. We have mentioned the novel use report(s) from the survey area and compared it with already reported plant rather than comparing the plant part(s) used. The literature search works with plant name and hence it was considered less likely that a previous report is overlooked because of different part use information. However, we have carefully rechecked our novelty information and found that novelty of plant species was intact at least up to the part use reported here for a specific disorder

3 R1 No detailed information has been given on how the plants were collected; whether they were used in fresh or dried state; if dry then how long can the medicinal activities of a given plant be sustained. The plants were collected in fresh form initially. Some of the plants were used in fresh form and some in dried form. Information incorporated in supplementary file 1. Mostly locals use the dried plants within 1-2 years of duration.

4 R1 Preparation of medicine like powder, decoction or other forms, dosage, part of plant used, use of monoherbal or polyherbal formulation. Detailed Information regarding preparation method is given in Supplementary file 1 (Columns JKL& M) noteworthy information is also mentioned in figure 3(a). Regarding formulation like monoherbal and polyherbal, information mentioned in supplementary file 1.

5 R1 If the same plant is used for more than one gastrointestinal disorder like say diarrhea and constipation did the mode of use differed and how? Mode of use of same plant for more than one gastrointestinal disorder also mentioned in supplementary file 1. The mode of use is same for more than one disease treated as mentioned in supplementary file 1.

6 R1 Endangerment of these medicinal plants and any efforts by the locals in their conservation. As such locals do not practice any conservation efforts however local people understand that excessive use of plant part(s) can lead the plant for its endangerment, so the locals tend to reduce the overhunting of plant part(s).

7 R1 The altitude range where the plants can be found. The altitude range where the plants were collected is mentioned in Fig 1, which between 35°23'14.3"N latitude and 74°34'44.7"E longitude, at an elevation range between 934-7937 m above the sea level.

8 R1 The authors note that elderly people possessed more knowledge of traditional plant usage. In that case, why were more elderly people not consulted to gather more information? That elderly people know more of their traditional customs is open knowledge; the question is how can the young generation be made more interested in the traditional customs? The population in the study area is not so high and we know that number of elders in a population is always comparatively low. We tried to consult the elders of the survey area aged between 50-60 that comprised up of 27.1% and above 60 years 25.9%. Secondly, in the study area some of the elders don’t want to share their traditional knowledge as they think that their traditional knowledge may be stolen by others. Thirdly, there is health factor of the elders also involved, as the elders are weaker than the youngsters, so they don’t have much energy to discuss with the researcher for a longer discussion.

 Response to reviewer 2 

1 R2 � The abstract is well-structured, well written and has excellently discussed and concluded the different aspects. However, it would be highly appreciated if authors incorporate sentence about the future perspectives of the research work conducted from present study. Plant species with maximum medicinal values could be a potential source of novel drug leads to cure gastrointestinal disorders.

2 R2 � Add the word “The” before “resent study was conducted in fairy meadows” in the abstract. Word “The” added as directed.

3 R2 � There is no future perspective statement in the abstract section. Provide a statement that what next can be done after these ethnobotanical surveys. Two sentences added at the end of the abstract which are “This showed that occupants of the study area have sound information about ethno-pharmacological consumption of medicinal plants with some novel use reports which may provide the basic data for further pharmacological research. Plant species with maximum medicinal values could be a potential source of novel drug leads to cure gastrointestinal disorders” shows the future perspective statement in the abstract section.

4 R2 � The introduction section is quite informative and to the point. Authors have comprehensively discussed the notion behind present research work. Authors are advised once again to double check grammar, sentence structure etc, if there any deficiency, fix them accordingly. All done. 

5 R2 � Provide updated literature to the statement “According to world health organization (WHO) estimates…………. million deaths on 2012 alone. Update 2021-2022 statistics if available. Updated as suggested. 

6 R2 � There is no discussion on the overall medicinal plants flora of the said geographical area. Provide few words and discuss about the dominant medicinal plants.

 Discussion added as suggested 

7 R2 � What was ecological distribution and phytogeography, utilization status and threats to the surveyed medicinal plants? Do you have some recommendations regarding their conservation aspects? I think you must add 1-2 lines. Added in results a separate subheading as “Phytogeographic distribution and utilization status of medicinal plants” recommendations regarding conservation of medicinal flora added in the conclusion 

8 R2 � Results and discussion section is well written, all results have been rightly discussed with relevant data from literature. Add some new 2020-2021 data from literature if available. Done as suggested 

9 R2 � The conclusion section is comprehensive and well-articulated. Remove results and introductory statements from conclusion section after carefully reviewing manuscript. Conclusion revised according to the suggestions

10 R2 � There is no uniformity in references. Cross check all the references and strictly follow PLOS ONE author guidelines. All done.

 Response to reviewer 3 

1 R3 Authors must concentrate on the formatting, and use of symbols, etc., throughout the manuscript. All done.

2 R3 Continuous line numbers are needed for point out the syntax error, corrections in the manuscript. Done

3 R3 Keywords should be unbold. Done.

4 R3 In introduction, authors have cited the WHO report 2017 it should be referred and cite after 2019 WHO report is better. Done as suggested 

5 R3 Authors should refer few more GIDs articles and include some lines in the introduction section. Because the GIDs information is looks shallow. Better to discuss deeply. Updated as suggested 

6 R3 Institutional ethical committee number should be added in the statement of ethics section. Ethical committee number added as UPR/HACE/01/13/21. Dated March 16, 2021. 

7 R3 Remove the dot in front of lines in Documentation of Ethnomedicinal Data section. Done

8 R3 In sub-title of Results and Discussion section should be “Results”. Because authors have written the separate discussion section. Done

9 R3 Framework figure is required. It will be useful to the readers for better understanding of the studied issue. Framework figure added with the map of the study area (Fig.1). As there is a limit of 5 figures from PLOS at max. 

10 R3 In materials and method section: authors should explain why each item of methodology was done. Explained 

11 R3 Proper formatting needed in the materials and methods, results, discussion, and conclusion section. Done.

12 R3 In conclusion section authors should provide some future prospectus related to present study. Done as suggested 

13 R3 Reference section should be formatted according to the journal. All done.

14 R3 Authors should clearly state the figure caption as well as description for ease of understanding of the readers. All done.

---

## [Decision Letter · Decision Letter 1]

23 May 2022

Ethnomedicinal appraisal of Plants used for the treatment of gastrointestinal complaints by tribal communities living in Diamir district, Western Himalayas, Pakistan

PONE-D-22-02797R1

Dear Dr. Khan,

We’re pleased to inform you that your manuscript has been judged scientifically suitable for publication and will be formally accepted for publication once it meets all outstanding technical requirements.

Kind regards,

Jen-Tsung Chen, Ph.D.

Academic Editor

PLOS ONE

Additional Editor Comments (optional):

Reviewers' comments:

Reviewer's Responses to Questions

**Comments to the Author**

1. If the authors have adequately addressed your comments raised in a previous round of review and you feel that this manuscript is now acceptable for publication, you may indicate that here to bypass the “Comments to the Author” section, enter your conflict of interest statement in the “Confidential to Editor” section, and submit your "Accept" recommendation.

Reviewer #1: All comments have been addressed

2. Is the manuscript technically sound, and do the data support the conclusions?

Reviewer #1: Yes

3. Has the statistical analysis been performed appropriately and rigorously? 

Reviewer #1: N/A

4. Have the authors made all data underlying the findings in their manuscript fully available?

Reviewer #1: Yes

5. Is the manuscript presented in an intelligible fashion and written in standard English?

Reviewer #1: Yes

6. Review Comments to the Author

Reviewer #1: Authors have answered all my previous comments.

7. PLOS authors have the option to publish the peer review history of their article (what does this mean?). If published, this will include your full peer review and any attached files.

Reviewer #1: **Yes: **Mohammed Rahmatullah

---

## [Editor Report · Acceptance letter]

30 May 2022

PONE-D-22-02797R1 

Ethnomedicinal appraisal of Plants used for the treatment of gastrointestinal complaints by tribal communities living in Diamir district, Western Himalayas, Pakistan 

Dear Dr. Khan:

I'm pleased to inform you that your manuscript has been deemed suitable for publication in PLOS ONE. Congratulations! Your manuscript is now with our production department. 

Kind regards, 

on behalf of

Dr. Jen-Tsung Chen 

Academic Editor

PLOS ONE